# Adenosine 2A Receptor Activation Amplifies Ibrutinib Antiplatelet Effect; Implications in Chronic Lymphocytic Leukemia

**DOI:** 10.3390/cancers14235750

**Published:** 2022-11-23

**Authors:** Omar Elaskalani, Grace Gilmore, Madison Hagger, Ross I. Baker, Pat Metharom

**Affiliations:** 1Telethon Kids Institute, Cancer Centre, Nedlands, WA 6009, Australia; 2Centre for Child Health Research, University of Western Australia, Crawley, WA 6009, Australia; 3Perth Blood Institute (PBI), Perth, WA 6005, Australia; 4Western Australian Centre for Thrombosis and Haemostasis (WACTH), Health Futures Institute, Murdoch University, Murdoch, WA 6150, Australia; 5Platelet Research Laboratory, School of Pharmacy and Biomedical Sciences, Curtin Health and Innovation Research Institute (CHIRI), Faculty of Health Sciences, Curtin University, Bentley, WA 6102, Australia

**Keywords:** CLL, platelets, CD73, adenosine, ibrutinib

## Abstract

**Simple Summary:**

A significant number of patients with chronic lymphocytic leukemia (CLL) have an increased risk of bleeding. This risk is further increased when taking ibrutinib, a new effective therapy for CLL. Platelets are the key player in haemostasis and thrombosis. In this study, we first characterized platelet function in untreated stable CLL patients in comparison to age-matched healthy volunteers. Using ex vivo platelets from healthy volunteers, we then investigated a possible mechanism of platelet dysfunction induced by a combination of a CLL-derived factor (adenosine) and ibrutinib. We found that therapeutic concentration of ibrutinib did not affect platelet activation response to collagen. However, the presence of adenosine switched off a central platelet activation pathway leading to increased antiplatelet activity of ibrutinib. Larger studies are needed to draw a correlation between adenosine, platelet function and ibrutinib-associated bleeding in CLL patients.

**Abstract:**

Chronic lymphocytic leukemia patients have an increased bleeding risk with the introduction of Bruton tyrosine kinase (BTK) inhibitors. BTK is a signaling effector downstream of the platelet GPVI receptor. Innate platelet dysfunction in CLL patients and the contribution of the leukemia microenvironment to the anti-platelet effect of BTK inhibitors are still not well defined. Herein, we investigated platelet function in stable, untreated CLL patients in comparison to age-matched healthy subjects as control. Secondly, we proposed a novel mechanism of platelet dysfunction via the adenosinergic pathway during BTK inhibitor therapy. Our data indicate that the nucleotidase that produces adenosine, CD73, was expressed on one-third of B-cells in CLL patients. Inhibition of CD73 improved platelet response to ADP in the blood of CLL patients ex vivo. Using healthy platelets, we show that adenosine 2A (A2A) receptor activation amplifies the anti-platelet effect of ibrutinib (10 nM). Ibrutinib plus an A2A agonist—but not ibrutinib as a single agent—significantly inhibited collagen (10 µg/mL)-induced platelet aggregation. Mechanistically, A2A activation attenuated collagen-mediated inhibition of p-VASP and synergized with ibrutinib to inhibit the phosphorylation of AKT, ERK and SYK kinases. This manuscript highlights the potential role of adenosine generated by the microenvironment in ibrutinib-associated bleeding in CLL patients.

## 1. Introduction

Chronic lymphocytic leukemia (CLL) represents the most common type of adult leukemia in Western countries. CLL is a gradual clonal expansion of mature B-cells in peripheral blood, bone marrow and secondary lymphoid organs caused by constitutive B-cell receptor signalling and maintained via overexpression of anti-apoptotic protein B-cell lymphoma 2 (BCL2), which results in a high lymphocyte count in circulating blood [1,2,3,4,5]. The B-cells produced are mature but non-functional, ultimately compromising the immune system [2].

Recently, oral Bruton tyrosine kinase (BTK) inhibitors have revolutionised the landscape of CLL treatment. BTK is a member of the TEC family tyrosine kinases and plays a crucial role in B-cell receptor (BCR) signal transduction. Upon ligand binding, BTK is activated, resulting in the increased expression of cell proliferation and survival genes [3]. In addition to B-cells, BTK plays a crucial role in platelet activation. Platelets are the key player in thrombosis and haemostasis. BTK is involved in von Willebrand factor (VWF) signalling and collagen-induced platelet aggregation [6].

Ibrutinib is a first-in-class, highly potent small molecule that irreversibly and covalently binds to the Cys481 residue within the ATP binding pocket of BTK [3,7,8,9]. Ibrutinib is a highly effective therapy for CLL and is prescribed indefinitely to achieve a durable response. However, adverse effects from ibrutinib can lead to discontinuation, which may lead to disease progression [10,11]. Bleeding has been reported in up to 50% of CLL patients using ibrutinib. The general pattern of bleeding is consistent with primary haemostatic failure (subcutaneous bleeding or bruising following minimal trauma) and excessive surgical haemorrhage [12,13]. This bleeding occurs despite improved platelet counts during ibrutinib treatment, suggesting that bleeding is a result of platelet dysfunction [14].

It is hypothesised that the high incidence of bleeding associated with ibrutinib use in CLL is due to a pre-existing and under-recognised mild platelet function defect exacerbated by platelet BTK inhibition [15]. Indeed, ibrutinib can inhibit platelet BTK/TEC kinases which are required for GPVI signalling [16]. GPVI is the primary collagen receptor in platelets. However, patients with a congenital defect in platelet GPVI or BTK do not present with excessive bleeding or life-threatening haemorrhage because redundancy of platelet activation pathways makes GPVI/BTK dispensable for life [17]. Moreover, bleeding has not been reported in healthy subjects taking ibrutinib [18,19]. These observations led us to propose that a factor or factors upregulated during CLL may broaden the anti-platelet spectrum of ibrutinib to inhibit multiple signalling pathways in platelets, exacerbating bleeding issues.

CLL cells and their microenvironment can generate adenosine, which is well known to be immunosuppressive and an inhibitor of platelet activity [20,21]. Unlike normal B-cells, CLL cells express hypoxia-inducible factor 1 (HIF1- α) even under normoxia. Hypoxia refers to a condition in which oxygen is limited and is governed by HIF1- α transcription factor [22]. Under hypoxic conditions, extracellular ATP and ADP are accumulated, and CD73 expression is increased via HIF1- α [23,24]. Different immune cells in the tumour microenvironment also express CD39 and CD73 and can also secrete ATP and ADP [25]. Adenosine is the end-product of the ectoenzymes CD39 (ATP → ADP → AMP) and CD73 (AMP → Adenosine). Platelets express adenosine receptors 2a and 2b (A2a, A2b). Adenosine can reduce platelet activation via activating protein kinase A (PKA)-mediated phosphorylation of vasodilator-stimulated phosphoprotein (VASP) and inhibition of calcium release [20].

The effect of the CLL-adenosine-rich microenvironment on platelet function is unknown. Herein, we show that the presence of adenosine and ibrutinib “switch off” several platelet activation pathways which could lead to an increased risk of bleeding in CLL patients. Therefore, targeting CD73 in CLL could be an additional strategy to offer a protection against the most common adverse reactions from BTK inhibitors. Targeting CD73 may also improve host immune response and slow disease progression [21,26].

## 2. Materials and Methods

### 2.1. Reagents

Collagen, ADP, ristocetin and epinephrine were obtained from Helena Labs. Australia, and CRP was a kind gift from Professor Elizabeth Gardiner, Australian National University. Ibrutinib and Acalabrutinib were obtained from Apexbio, USA. HENECA was obtained from Abcam, UK. PSB-12379 was obtained from MedChemExpress, USA. Western blots antibodies were purchased from Cell Signalling Technology, USA. Flow cytometry antibodies were obtained from BD Biosciences, USA. Mepacrine was obtained from SigmaAldrich, USA.

### 2.2. Preparation of Human Platelets

Blood was collected from CLL patients or healthy controls in acid-citrate-dextrose (for washed platelets preparation) or in Na Citrate (for Platelet Rich Plasma, PRP) tubes with written informed consent as required by the Curtin University Human Research Ethics Committee, approval number: HRE2020-0384. After collection, washed platelets were prepared as described in previous literature [27,28]. Briefly, blood was centrifuged for 20 min at 150× *g*, after which PRP was carefully collected and then centrifuged for 10 min at 800× *g*. The supernatant was discarded, and the platelet pellet was resuspended in CGS buffer (33.33 mM glucose, 123.2 mM NaCl and 14.7 mM trisodium citrate, pH 7) and then centrifuged for another 10 min at 800× *g*. This was repeated for a total of 3 washes. Prostaglandin E1 (PGE1, 1 µM) was added before each centrifugation to prevent early activation of the platelets. The platelets were then resuspended in calcium-free Tyrode buffer (5.5 mM glucose, 12 mM NaHCO3, 0.49 mM MgCl2, 2.6 mM KCl, 5 mM HEPES, 0.36 mM NaH2PO4 and 138 mM NaCl, pH 7.4) at a concentration of 1 × 10^9^/mL. Prior to experimentation, the platelets were supplemented with 1.8 mM CaCl2 to provide a better environment for platelet activation for the aggregation assay.

For PRP preparation, citrated blood was centrifuged at 200 rpm for 5 min. The PRP was removed, and a platelet count preformed. PRP was diluted to 250 × 10^9^/mL using Platelet Poor Plasma from the patient. PPP was obtained by centrifugation at 2000 rpm for 15 min.

### 2.3. Light Transmission Aggregometry (LTA)

The responses of PRP or washed platelets from CLL patients or age-matched healthy volunteers to varying concentrations of agonists including collagen, ADP, ristocetin and epinephrine were assessed using a light transmission aggregometer (LTA, AggRAM^®^ Helena). The aggregation reaction proceeded for 10 min at 37 °C with constant stirring at 600 rpm.

For BTK inhibitor experiments, washed platelets from healthy volunteers at a concentration of 3 × 10^8^/mL were incubated in Tyrode-HEPES buffer supplemented with 1.8 mM CaCl2 with either PBS (vehicle control), ibrutinib (10 nM) or acalabrutinib (20 or 50 nM) for 1 h. Where HENECA was used, the reactions were incubated for 55 min, then HENECA was added 5 min before the agonist (either collagen 10 µg/mL or CRP 10 and 50 µg/mL), after which the platelet aggregation was measured. Platelet aggregation was recorded for at least 6 min using a light transmission aggregometer (Model 700 Aggregometer, Chrono-Log Corporation, Havertown, PA, USA) at 37 °C with constant stirring at 600 rpm.

### 2.4. Platelet Function Assay in Whole Blood

Platelet function assay was conducted in citrated blood using multiple electrode aggregometry (MEA—Multiplate^®^, Haemoview) as per manufacturer’s protocol [29].

### 2.5. Platelet Activation and GPVI Expression Using Flow cytometry

PRP was stained with mouse anti-human CD42b-PE antibody or mouse anti-human GPVI-Alexa Fluor^®^ 647 antibody (clone HY101) or relevant isotype controls for 15 min in the dark. For platelet activation in whole blood, citrated blood was diluted 1:50 in Tyrode-HEPES buffer and incubated with the inhibitors (CD73 inhibitors: PSB12379 100 µM or anti-CD73 mAb 1:200) for one h at 37 °C. ADP (2 µM) or PBS was added, and the samples were incubated for a further 15 min at 37 °C. The treated samples were single-, double- or triple-stained with mouse anti-human CD41a-BV421 antibody, mouse anti-human CD62P (P-Selectin)-PE antibody or mouse anti-human PAC1 (active form of αIIbβ3)-FITC antibody, or relevant isotype control antibodies for 15 min in the dark. All samples (PRP or blood) were fixed in ice-cold 1% paraformaldehyde in PBS and analysed by the Gallios flow cytometer (Beckman).

### 2.6. Platelet Mepacrine Uptake and Release Assay Using Flow Cytometry

Platelet dense granule content uptake and release were examined using mepacrine assay as previously described [30]. Briefly, freshly collected citrated blood from CLL patients or age-matched healthy volunteers were diluted 1:40 using HBSS buffer (without Ca or Mg). Forty µl of blood was labelled with mepacrine (5 µL, 2µM final concentration) and mouse anti-human CD41a-BV421 for 20 min at 37 °C. Thrombin 0.1 U/mL or buffer was then added and the samples were incubated for another 5 min. Samples were then diluted with 500 µl of PBS and immediately analysed with flow cytometry Gallios flow cytometer (Beckman).

### 2.7. ELISA

Citrated blood from CLL patients or age-matched healthy volunteers was used to prepare plasma which was aliquoted and stored at −80 °C until further analysis. The levels of sCD73 were measured using a Human CD73 ELISA Kit (NT5E) (ab213761) according to the manufacturer’s instructions. The concentrations of sCD73 concentrations in the samples were calculated from a standard curve generated at 450 nm using a plate reader (EnSpire Multimode, PerkinElmer^®^).

### 2.8. Western Blot

The antibodies used were all rabbit antibodies, specific for p-SYK (Tyr 352), p-BTK (Tyr 223), p-PLCγ2 (Tyr 759), p-AKT (Ser 473), p-VASP (Ser 239), p-ERK1/2 (Thr 202/ Tyr 204) and α-actinin. After the aggregation assay, the reactions were lysed using LaemmLi sample buffer (Bio-RAD) in combination with Protease/Phosphatase Inhibitor Cocktail (Cell Signalling Technology) and β-mercaptoethanol. The samples were then heated at 95 °C for 5 min. Samples (40 µl) were loaded in each lane and separated using sodium dodecyl sulphate-polyacrylamide gel electrophoresis. After separation, the proteins were transferred to a polyvinylidene difluoride (PVDF) membrane. Non-specific binding was then blocked by covering the membrane with 5% non-fat powdered milk in Tris-buffered saline with 0.1% Tween 20 (TBS-T) at room temperature for 30 min. After washing the membrane with TBS-T 3× for 5 min each, the relevant antibody (at a dilution of 1:1000) was added to the membrane, which was then left to incubate overnight at 4 °C. The secondary antibody, horseradish peroxidase-conjugated anti-rabbit antibody at a dilution of 1:20,000 (Jackson Immune Research, West Grove, PA, USA), was used to detect the primary antibody. After the secondary antibody was left to incubate for 3 hs at room temperature and washing the membrane in TBS-T buffer, Amersham ECL Western Blotting Detection Reagent (GE Healthcare Life Sciences, Chicago, IL, USA) was used to develop the membrane, and the resulting chemiluminescence was recorded using the ChemiDoc imaging system (Bio-Rad, Hercules, CA, USA).

### 2.9. Statistical Analysis

Data were analysed using GraphPad PRISM 9.0 software (GraphPad Software). Results are expressed as the mean ± standard error (SEM). Student’s *t*-test or one-way ANOVA were used to examine the significance of the mean as appropriate and as indicated in the figure legend. Any differences found were considered statistically significant at *p* < 0.05.

## 3. Results

### 3.1. Low GPVI Expression and Collagen Response in CLL Patients

In this study, we obtained fresh blood samples from stable untreated CLL patients with normal platelet counts (>100 × 109/L) (n = 23) and age-matched healthy volunteers (n = 11). None of the study participants had been using any antiplatelet medications in the previous seven days. GPVI is the main collagen receptor in platelets. Ibrutinib inhibits BTK downstream of GPVI [31]. Therefore, we thought to examine GPVI expression and response to collagen in ibrutinib-naïve CLL patients. GPVI expression in PRP was examined by flow cytometry. Our data (Figure 1A) indicate that CLL patients express less GPVI compared to age-matched healthy volunteers. Collagen response was significantly impaired in whole blood and PRP (Figure 1B–D). Plasma soluble factors and CLL cells can affect platelet function [26]. Therefore, we examined platelet response to collagen in washed platelets. CLL platelet response to collagen 1 µg/mL but not 5 µg/mL was impaired (*p* = 0.03 and 0.097, respectively).

### 3.2. Broad Disruption of Platelet Response in CLL Patients

Next, we examined platelet response to several agonists that cover various platelet activation pathways. Platelet responses to ristocetin, epinephrine and ADP were impaired in PRP (Figure 2). In whole blood multiplate assays, platelet response was impaired to ADP but not epinephrine (*p* = 0.056). Mepacrine uptake was reduced in CLL patients indicating a general storage pool disorder which may explain the broad disruption of platelet response to several agonists that work by different mechanisms [30].

### 3.3. CD73 Contributes to Platelet Function in CLL Patients

CD73 is the rate-limiting enzyme in the production of adenosine from ATP. Previous studies have reported an increase in CD73 expression in solid tumours and CLL cells. CD73 expression and excess production of adenosine is thought to be induced by hypoxia [23,25]. The role of CD73 in cancer progression and tumour-immune tolerance is well established, and several clinical trials are currently testing the combination of CD73 inhibition with chemo- or immunotherapy in several types of cancer [21]. However, adenosine generated by CD73 is also a well-established platelet inhibitor [20]. Therefore, we examined if CD73-generated adenosine was relevant to platelet hypofunction in CLL patients. To answer this question, we examined CD73 expression in peripheral blood mononuclear cells (PBMCs) obtained from stable, untreated CLL patients and age-matched healthy volunteers. As previously described [23], although the expression level of CD73 per B-cells in most CLL patients is less than the controls (Figure 3B), the actual number of CD73+CD19+ cells in PBMCs are increased due to the high numbers of B-cells in CLL patients (Figure 3A). We next confirmed the presence of soluble CD73 in the plasma of CLL patients (Figure 3C). Using a selective inhibitor of CD73 activity (PSB12379, 10 µM) [32], we show that inhibition of CD73 in the whole blood of CLL patients increased ADP-induced platelet activation as measured by the increase in active αIIbβ3 expression (Figure 3D). These data show for the first time a potential role of CD73-generated adenosine in platelet dysfunction in CLL patients.

### 3.4. Ibrutinib Inhibits CRP but Not Collagen-Induced Platelet Aggregation

In order to understand the relevance of CD73-generated adenosine to the increased risk of bleeding reported with the use of BTK inhibitors, we first characterised their effect, at clinically relevant concentration, on platelet response to CRP and collagen (Fig. 4). Ibrutinib or acalabrutinib was incubated with washed platelets for 1 h at 37 °C, then CRP (10 or 50 µg/mL) or collagen (10 µg/mL) was added. Ibrutinib but not acalabrutinib completely abrogated CRP (10 µg/mL)-induced platelet aggregation (Figure 4A). However, at higher CRP concentration (50 µg/mL), ibrutinib still managed to significantly decrease platelet aggregation, whereas acalabrutinib was not effective (Figure 4B). Both agents had no effect on collagen-induced platelet aggregation (Figure 4A). We next examined the signalling kinases (PLCγ and BTK) downstream of GPVI receptor, the main CRP and collagen receptor in platelets [17]. Ibrutinib inhibited both CRP-induced PLCγ and BTK phosphorylation, which translated to a reduced platelet aggregation. In response to collagen, ibrutinib completely inhibited BTK but not PLCγ phosphorylation. Acalabrutinib (20 nM) reduced collagen (10 µg/mL) but not CRP (50 µg/mL)-induced phosphorylation of PLCγ and BTK (Figure 4C).

### 3.5. A2A Activation Amplifies the Antiplatelet Effect of Ibrutinib

Previous studies have reported reduced collagen-induced platelet aggregation in ibrutinib-treated patients [14]. However, our mechanistic studies here show that clinically relevant concentration of ibrutinib (10 nM) did not inhibit collagen-induced platelet aggregation, despite reducing BTK phosphorylation. Therefore, we hypothesized that local adenosine generated by CD73 in CLL patients could amplify the effect of ibrutinib to inhibit collagen-induced platelet aggregation. In order to test this hypothesis, we used a more stable derivative of adenosine (HENECA) +/− ibrutinib. We found that HENECA significantly potentiated the effect of ibrutinib and acalabrutinib (Figure 5A,B) to inhibit collagen-induced platelet aggregation. Mechanistically, the addition of HENECA reduced collagen plus ibrutinib or acalarbrutinib-induced Akt, Erk and Syk activation, while increasing VASP phosphorylation, indicating a broad inhibition of platelet activation (Figure 5C,D).

## 4. Discussion

In this study, we characterised platelet dysfunction in stable, untreated CLL patients at baseline. Since platelets become hyperactive with ageing, and most CLL patients are above 60 years old, we used age-matched healthy volunteers as controls to CLL patients [33]. CLL patients (n = 23) had normal platelet count and their plasma level of von Willebrand factor was comparable to healthy controls (Table 1 and Appendix A). CLL patients displayed lower platelet responses to collagen, ristocetin, epinephrine and ADP as measured by platelet aggregation and activation assays using LTA, MEA, and flow cytometry (Figure 1 and Appendix A). This broad inhibition of platelet responses is most likely due to the CLL platelets having less GPVI expression and dense granules (Figure 1 and Figure 2). Low platelet count is the most reliable marker to discriminate CLL patients with a high risk of bleeding (8). However, this study is the first to describe platelet function in CLL patients with normal platelet count in comparison to age-matched healthy controls.

It has been previously suggested that CLL cells could reduce platelet activation [34]. More recently, it has been shown that platelet response to collagen improved with the reduction of lymphocyte count in CLL patients [8]. Therefore, we proposed that CLL-related factors can contribute to platelet function at baseline and potentiate the antiplatelet effect of ibrutinib. The CLL microenvironment is rich in CD73-generated adenosine, which can inhibit platelet function [20,23]. In this study, we confirmed previous reports that showed the abundance of CD73+/CD19+ lymphocytes in CLL patients [23]. Moreover, we showed the presence of soluble CD73 in CLL plasma. Using a selective CD73 small-molecule inhibitor, we showed that inhibition of CD73 increased ADP-induced αIIbβ3 expression in platelets from CLL patients (Figure 3). However, this inhibition was not enough to reverse the low platelet aggregation in CLL patients as measured by MEA assay (data not shown).

Ibrutinib use in CLL patients is associated with bleeding risk, previously described as a result of the antiplatelet effect of the drug. However, individuals with a congenital defect in BTK or its upstream target GPVI do not present significant bleeding issues [17], suggesting involvement of other antiplatelet factors in CLL. In our study, we report for the first time that a clinically relevant concentration of ibrutinib (10 nM) [35] and acalabrutinib (50 nM) [36] can only inhibit CRP but not collagen-induced platelet aggregation (Figure 4). CRP is a specific agonist for GPVI/BTK pathway, unlike collagen which also engages the integrin receptor α2β1 [37]. This result confirms that platelet dysfunction in CLL patients with ibrutinib is not a result of BTK or GPVI inhibition alone. Previous clinical data also showed reduced CRP and collagen mediated platelet aggregation in ibrutinib-treated CLL patients, while healthy individuals did not experience bleeding issues with ibrutinib [14,31].

Several studies have examined the antiplatelet effects of micromolar concentration (0.5–10 µM) of ibrutinib and acalabrutinib in vitro on washed platelets [6,16]. However, ibrutinib and acalabrutinib are highly bound to the plasma proteins (97–98%); therefore they are physiologically unlikely to have “free, unbound” concentrations at the micromolar range for such drugs with these physiochemical properties. According to FDA reports, the free unbound maximum plasma concentrations of ibrutinib and acalabrutinib achieved in CLL patients at steady state are at the lower nanomolar range [35,36]. Based on studies in cancer cells, these concentrations (≤20 nM) are sufficient to inhibit BTK in vitro (and in vivo) [38,39,40,41]. Herein, we used the same incubation time (1 h) and drug concentrations previously used in BTK inhibition studies in cancer cells, and reflected the exposure of platelets to the pharmacologically active fractions of the drugs in vivo. In these conditions, ibrutinib and acalabrutinib inhibited CRP-mediated platelet aggregation. The effect of ibrutinib but not acalabrutinib persisted at high CRP concentration (50 µg/mL), reflecting the previously known superior potency of ibrutinib over acalabrutinib as a BTK inhibitor [42]. Both ibrutinib and acalabrutinib reduced collagen-induced BTK activation even in the absence of functional inhibition of platelet aggregation (Figure 4C).

Collagen-induced platelet aggregation is inhibited in ibrutinib-treated patients [14], although clinically relevant concentrations of ibrutinib lack an effect on collagen-induced platelet aggregation as indicated in our study. This in vitro–in vivo discrepancy may be explained by the presence of a disease-related biological factor which may synergise with ibrutinib to inhibit collagen-induced platelet aggregation. Based on several studies, the CLL microenvironment is rich in adenosine. In this manuscript, we propose that adenosine can amplify the antiplatelet effects of ibrutinib (Figure 5). The addition of the stable adenosine derivative HENECA [43] to ibrutinib resulted in significant inhibition of collagen-induced platelet aggregation. This inhibition was characterised by an increase in phosphorylation of VASP and a reduction in AKT, ERK and SYK phosphorylation. These changes highlight a broad inhibition in platelet activity, most likely caused by an adenosine-mediated increase in cyclic-AMP and protein kinase A, upstream effectors of VASP phosphorylation, resulting in reduced platelet activity [20]. VASP regulates actin filaments dynamics and filopodia formation during platelet activations. Increased platelet p-VASP negatively correlates with platelet aggregation and the activation of the integrin receptor α2bβ3 [44,45].

The correlation between plasma adenosine level and platelet activation has been previously established in many diseases [46,47,48,49,50,51]. Moreover, there has been an increasing interest in targeting the adenosine receptor A2 or CD73 in cancer immunotherapy [21]. However, the impact of adenosine on blood haemostasis in cancer has not been explored yet. Our study proposes a novel role of CD73/adenosine in platelet function in CLL patients. A larger study is required to draw a correlation between CD73, plasma adenosine level and platelet function in ibrutinib-treated and -untreated CLL patients. Such a study could provide much-needed data for a potential biomarker and a therapeutic target in CLL.

## 5. Conclusions

Our data suggest that multiple disease-related factors in addition to BTK inhibition are responsible for ibrutinib-associated bleeding in CLL patients. CLL patients on ibrutinib are typically elderly with multiple comorbidities and on other medications that may affect platelet functions (e.g., aspirin or fish oil). However, our study is the first to show that untreated, stable CLL patients, with normal platelet count, and not on any medications that affect platelet function, have lower platelet reactivity compared to age-matched healthy volunteers. We conclude that ibrutinib by itself is not sufficient to cause broad inhibition of platelet function or bleeding in CLL patients with normal platelet count. We propose, based on our results, that CLL-generated adenosine, plus platelet defects at baseline, synergize with ibrutinib to cause bleeding.

## Figures and Tables

**Figure 1 cancers-14-05750-f001:**
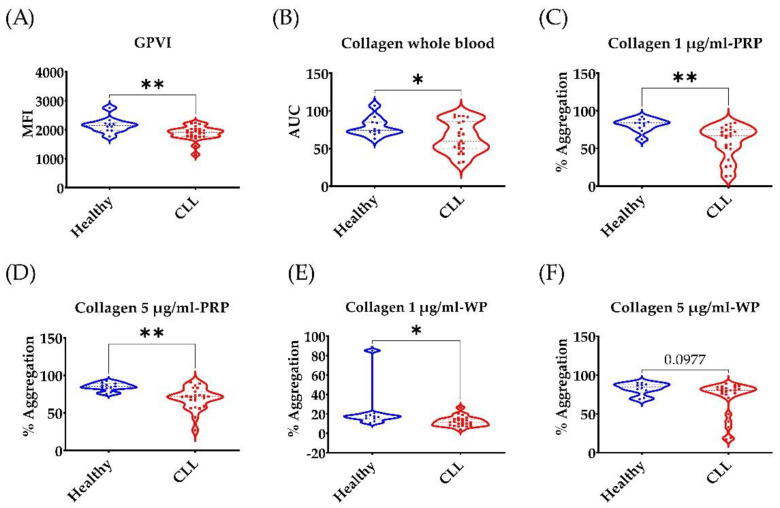
**Low GPVI expression and collagen response in CLL patients compared to age-matched healthy volunteers**. (**A**) Geometrical mean fluorescence intensity of GPVI-AF647 in PRP obtained from untreated CLL patients compared to age-matched healthy volunteers. (**B**–**F**) Platelet function analysis in whole blood (**B**), PRP (**C**,**D**) and washed platelets (WP) (**E**,**F**) in response collagen. * *p* < 0.05, ** *p* <  0.01; n  =  23 CLL patients and 11 healthy volunteers. Platelet aggregation in PRP was measured by LTA. Whole blood aggregation was measured by MEA using collagen 3.2 µg/mL.

**Figure 2 cancers-14-05750-f002:**
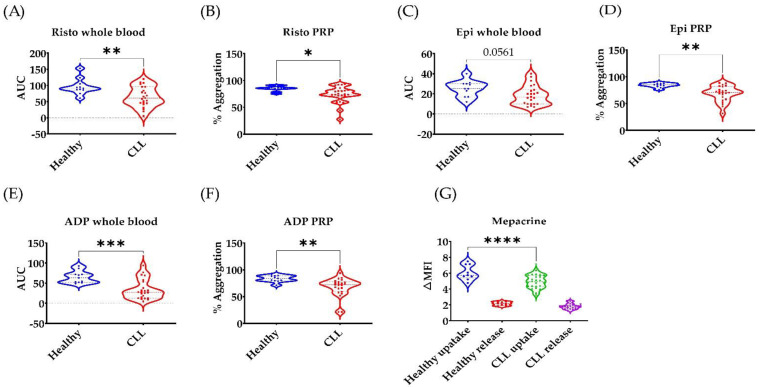
**Broad reduction in CLL platelet response to several agonists.** (**A**–**F**) Impaired platelet response to ristocetin, epinphrine and ADP in whole blood and PRP. (**G**) Reduced mepacrine uptake in CLL platelets compared to health volunteers. * *p* < 0 .05, ** *p*  <  0.01, *** *p*  <  0.001, **** *p*  <  0.0001; n  =  23 CLL patients and 11 age-matched healthy volunteers. Platelet aggregation in PRP was measured by LTA using ristocetin: 1.2 g/L, epinephrine: 7 µM, and ADP 4 µM. Whole blood aggregation was measured by MEA using risotcetin 0.7 g/L, ADP 6.4 µM and epinephrine 7 µM.

**Figure 3 cancers-14-05750-f003:**
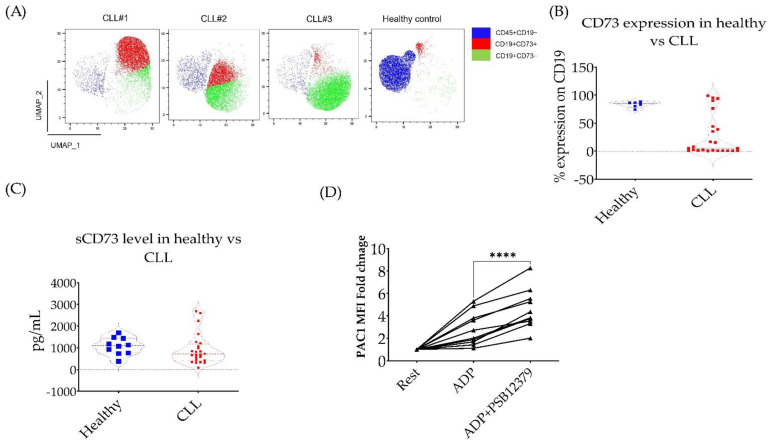
**Low platelet response to ADP is partially improved by CD73 inhibition.** (**A**) Representative UMAPs of CD73 expression in three different CLL patients and age-matched healthy control. (**B**) Combined data of CD73 expression on CD19 B-cells in the peripheral blood of CLL patients (red, n = 23) and age-matched healthy volunteers (blue, n = 6). (**C**) Soluble CD73 in the plasma of CLL patients (red, n = 23) and age-matched healthy volunteers (blue, n = 12). (**D**) The effect of CD73 inhibition on ADP-induced platelet activation in whole blood obtained from CLL patients, n = 10. Blood was incubated with the CD73 inhibitor (PSB12379. 10 µM) or vehicle control for 1 h, then ADP (2 µM) was added and platelets were further incubated for 15 min. Platelets’ activity was then assessed using flow cytometry and PAC1 antibody which only detects the activated platelet glycoprotein receptor αIIbβ3. UAMP (Uniform Manifold Approximation and Projection) was generated using FlowJo. **** *p* <  0.0001.

**Figure 4 cancers-14-05750-f004:**
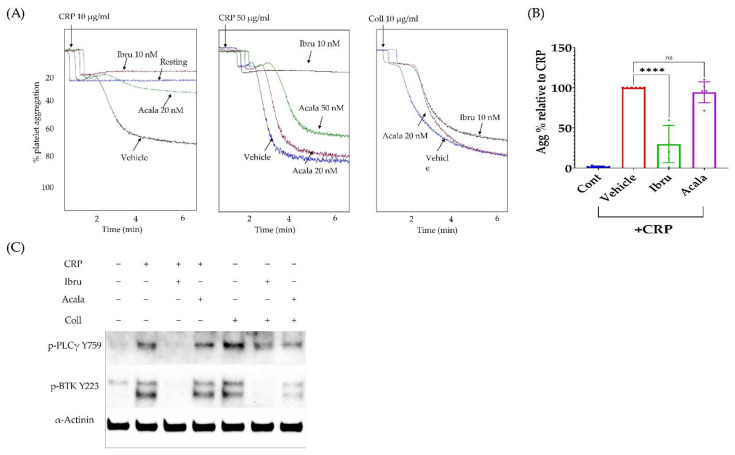
**Ibrutinib (10 nM) inhibits CRP but not collagen-induced platelet aggregation**. (**A**) Representative aggregation traces showing the effect of ibrutinib (10 nM) and acalabrutinib (20 nM) on CRP (10 and 50 ug/mL) and collagen (10 ug/mL)-induced platelet aggregation. Washed platelets were pre-incubated with the inhibitors for 60 min at 37 °C before addition of the agonist. (**B**) Combined results of ibrutinib (10 nM) and acalabrutinib effect (20 nM) on CRP (50 µg/mL)-induced platelet aggregation (n = 5). (**C**) The effect of ibrutinib and acalabrutinib on CRP (50 µg/mL)-and collagen (10 µg/mL) -induced activation of PLCγ2 and BTK phosphorylation. Ibru: ibrutinb; Acala: acalabrutinib. **** *p* < 0.0001; Insignificance is denoted by ns.

**Figure 5 cancers-14-05750-f005:**
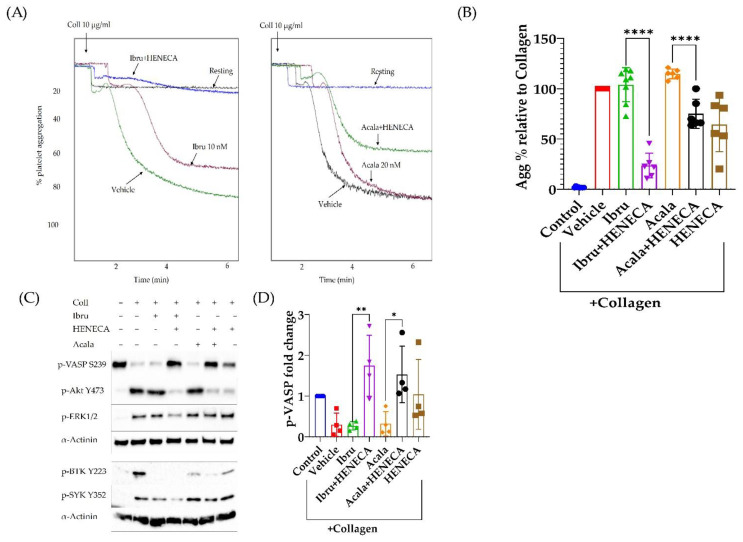
**Adenosine 2A receptor activation amplifies the antiplatelet effects of ibrutinib by inhibiting collagen-mediated changes in VASP phosphorylation.** (**A**) Representative aggregation traces showing the effect of ibrutinib (10 nM) and acalabrutinib (20 nM) ± HENECA (stable adenosine derivative, 10 uM) on collagen (10 ug/mL)-induced platelet aggregation. Washed platelets were pre-incubated with the inhibitors for 55 min and with HENECA for 5 min at 37 °C before addition of collagen. (**B**) Combined results of the effect of ibrutinib and acalabrutinib ± HENECA on collagen-induced platelet aggregation (n ≥ −5). (**C**) The effect of ibrutinib and acalabrutinib ± HENECA on collagen-induced changes in BTK, SYK, VASP, Akt and ERK phosphorylation. (**D**) Combined results of the effect of ibrutinib and acalabrutinib ± HENECA on VASP phosphorylation. Data are presented as fold change compared to control (n = 4). * *p* < 0 .05, ** *p* < 0.01, **** *p* < 0.0001.

**Table 1 cancers-14-05750-t001:** Demographic and haematological parameters of untreated CLL patients and age-matched healthy controls.

	Untreated CLL Cohort	Health Cohort	*p*-Value (CLL vs. Healthy)
**Number of participants**	23	11	
**Median Age, yr (range)**	69 (52–89)	62 (57–69)	0.0389
**WBC (** **×10^9^/L), mean (range)**	40.82 (13–112.3)	5.54 (4.1–7.9)	0.0004
**Platelets (** **×10^9^/L), mean (range)**	215.1 (128–327)	231.5 (138–326)	0.4247
**MPV (fL), mean (range)**	8.57 (7–10.2)	9.18 (8.2–10.4)	0.0813
**Lymphocytes (** **×10^9^/L), mean (range)**	34.49 (10.7–103.9)	1.63 (1–2.6)	0.0006
**Lymphocytes %, mean (range)**	79.91 (61–94.1)	29.47 (18.4–36.3)	<0.0001
**RBC (** **×10^12^/L), mean (range)**	4.72 (3.6–5.9)	5.00 (4.3–5.6)	0.1563
**Hb g/L, mean (range)**	137.0 (116–158)	146.3 (127–169)	0.0446
**HCT %, mean (range)**	41.89 (32.9–50.6)	44.88 (38.6–51.1)	0.0552

## Data Availability

The data presented in this study are available on request from the corresponding author.

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
