# Peer review of "Adenosine 2A Receptor Activation Amplifies Ibrutinib Antiplatelet Effect; Implications in Chronic Lymphocytic Leukemia"

_cancers, 2022, doi:10.3390/cancers14235750_

Round 1
Reviewer 1 Report
Journal: Cancers MDPI
Title: Adenosine 2A receptor activation amplifies ibrutinib antiplatelet effects; a potential role of CD73 in platelet defects in chronic lymphocytic leukemia
Type: Research article
Manuscript ID: cancers-2018628
Authors: Omar Elaskalani, Grace Gilmore , Madison Hagger , Ross I Baker , Pat Metharom
Description: The present manuscript reports on an investigation of the platelet function in stable, untreated Chronic lymphocytic leukemia affected patients in comparison to age matched healthy controls. Then, the authors proposed a novel mechanism of platelet dysfunction via the adenosinergic pathway during Bruton tyrosine kinase inhibitor therapy with ibrutinib. Main results indicate that the nucleotidase that produces adenosine, CD73, was expressed on one-third of B-cells in CLL patients. In addition, inhibition of CD73 improved platelet response to ADP in the blood of CLL patients ex vivo. Moreover, adenosine 2A (A2A) receptor activation has been described to amplify the anti-platelet effect of ibrutinib (10 nM). Authors concluded that their data suggest that multiple disease-related factors in addition to BTK inhibition are responsible for ibrutinib-associated bleeding in CLL patients.
Revision: The authors may consider the following points for improving the manuscript.
1) Line 30, instead of “matched healthy controls” I would say “matched healthy subjects, as control”
2) Line 50 BTK as well as other acronyms should be mentioned with their complete name the first time being mentioned. Please revise the manuscript for similar additional errors
3) Lines 58-59 references describing the therapeutic role of Ibrutinib on Chronic lymphocytic leukemia therapy should be included. For instance https://www.ncbi.nlm.nih.gov/pmc/articles/PMC7266824/
4) Lines 85-87 or 335-338 a very detailed description of the adenosine pathway and of the targeting of the adenosine signaling in cancer is reported here (https://www.nature.com/articles/s41388-021-02090-z). This reference should be included.
5) Please include more refs supporting the methods
6) Demographic and hematological parameters reported in table 1 should be statistically analyzed, while P value should be included.
7) For a better reading, all figure’s and subhead’s titles should not comprise acronyms but complete names
8) Al figures sohld be improved in terms of readability, several words are almost unreadable. E.g., figure 4 panel a.
9) Figures 4 and 5, panel A, underlining in red should be removed.
Author Response
1) Line 30, instead of “matched healthy controls” I would say “matched healthy subjects, as control”
Corrected
2) Line 50 BTK as well as other acronyms should be mentioned with their complete name the first time being mentioned. Please revise the manuscript for similar additional errors
Corrected
3) Lines 58-59 references describing the therapeutic role of Ibrutinib on Chronic lymphocytic leukemia therapy should be included. For instance https://www.ncbi.nlm.nih.gov/pmc/articles/PMC7266824/
References have been added
4) Lines 85-87 or 335-338 a very detailed description of the adenosine pathway and of the targeting of the adenosine signaling in cancer is reported here (https://www.nature.com/articles/s41388-021-02090-z). This reference should be included.
The reference has been added
5) Please include more refs supporting the methods
References have been added
6) Demographic and hematological parameters reported in table 1 should be statistically analyzed, while P value should be included.
Data were analysed and put as a supplementary figure 1. P-values are added in table 1
7) For a better reading, all figure’s and subhead’s titles should not comprise acronyms but complete names
Corrected
8) Al figures sohld be improved in terms of readability, several words are almost unreadable. E.g., figure 4 panel a.
9) Figures 4 and 5, panel A, underlining in red should be removed
All figures have been re-made for better readability
Reviewer 2 Report
This study is interesting with clinical significance. Taking ibrutinib increases the risk of bleeding on patients with chronic lymphocytic leukemia (CLL), which is a clinical problem that needs to be solved. The authors put forward a new point of view to solve this problem. The followings are comments to the authors.
1.I suggest that the title of the article need be rewritten, which is not clear.
2.In line 18, the first “;” should be “,” and the second “;”should be deleted.
3. The number of participants of untreated CLL Cohort is 23 in Table 1, but the number CLL patients is 24 in Figure 1 legend. Please confirm the number.
4. How did the authors choose the concentration of the drug in the study? For example, ibrutinib (10 nM) and acalabrutinib (20 nM).
5. There are no letters of A-C in Figure 4. Please add those.
Author Response
1.I suggest that the title of the article need be rewritten, which is not clear.
The title has been changed (Adenosine 2A receptor activation amplifies ibrutinib antiplatelet effect; implications in chronic lymphocytic leukemia)
2.In line 18, the first “;” should be “,” and the second “;”should be deleted.
Fixed
- The number of participants of untreated CLL Cohort is 23 in Table 1, but the number CLL patients is 24 in Figure 1 legend. Please confirm the number.
The number is 23, and has been fixed in the figure legends
- How did the authors choose the concentration of the drug in the study? For example, ibrutinib (10 nM) and acalabrutinib (20 nM).
From the FDA files of both drugs, references 37 and 38
Ibrutinib (FDA, 2013(205552Orig1s000)): In table 3, section 2.2.5.1, ibrutinib after continuous fixed dose of 8.3 mg/kg achieves a Cmax of 155 ± 126 ng/ml. Ibrutinib is 97.7% bound to the plasma protein, so the free, unbound pharmacologically active concentration is roughly 3.5 ± 2.9 ng/ml ~ 7.9 ± 6.5 nM. Therefore, we choose to use a fixed concentration of 10 nM.
Acalabrutinib (FDA, 2017(210259Orig1s000), the Cmax for reference subjects (white, male, 80 kg, with normal renal and hepatic function) is 323 (47.7-934) ng/ml. Considering the plasma protein binding of acalabrutinib is 98%, so the free, unbound pharmacologically active concentration is roughly 6.46 (0.95-18.68) ng/ml or ~ 13.8 (2-40.8) nM, therefore we used 20 and 50 nM
*The plasma concentration of these drugs will also be affected by food, drinks, drug interactions, renal and hepatic functions but it is unlikely to reach micromolar concentrations.
References for other inhibitors (PSB12379 and HENECA) have been added
- There are no letters of A-C in Figure 4. Please add those.
Letters have been added